# Factors associated with SARS-COV-2 positive test in Lifelines

**Grigory Sidorenkov**[1]*, **Judith M. Vonk**[1], **Marco Grzegorczyk**[2], **Francisco O. Cortés-Ibañez**[1], **Geertruida H. de Bock**[1]

**1** Department of Epidemiology, University of Groningen, University Medical Center Groningen, Groningen, The Netherlands, **2** Computer Science and Artificial Intelligence, University of Groningen—Bernoulli Institute for Mathematics, Groningen, Netherlands

* g.sidorenkov@umcg.nl

## Abstract

### Background

Severe acute respiratory syndrome coronavirus-2 (SARS-COV-2) can affect anyone, however, it is often mixed with other respiratory diseases. This study aimed to identify the factors associated with SARS-COV-2 positive test.

### Methods

Participants from the Northern Netherlands representative of the general population were included if filled in the questionnaire about well-being between June 2020-April 2021 and were tested for SARS-COV-2. The outcome was a self-reported test as measured by polymerase chain reaction. The data were collected on age, sex, household, smoking, alcohol use, physical activity, quality of life, fatigue, symptoms and medications use. Participants were matched on sex, age and the timing of their SARS-COV-2 tests maintaining a 1:4 ratio and classified into those with a positive and negative SARS-COV-2 using logistic regression. The performance of the model was compared with other machine-learning algorithms by the area under the receiving operating curve.

### Results

2564 (20%) of 12786 participants had a positive SARS-COV-2 test. The factors associated with a higher risk of SARS-COV-2 positive test in multivariate logistic regression were: contact with someone tested positive for SARS-COV-2, $\geq$1 household members, typical SARS-COV-2 symptoms, male gender and fatigue. The factors associated with a lower risk of SARS-COV-2 positive test were higher quality of life, inhaler use, runny nose, lower back pain, diarrhea, pain when breathing, sore throat, pain in neck, shoulder or arm, numbness or tingling, and stomach pain. The performance of the logistic models was comparable with that of random forest, support vector machine and gradient boosting machine.

**Data Availability Statement:** Data may be obtained from a third party and are not publicly available. Researchers can apply to use the Lifelines data used in this study. More information about how to request Lifelines data and the

conditions of use can be found on their website (https://www.lifelines.nl/researcher/how-to-apply).

**Funding:** The study was funded by the Department of Epidemiology, University Medical Center Groningen, the Netherlands. The funders had no role in study design, data collection and analysis, decision to publish, or preparation of the manuscript. The study used the data of The Lifelines Initiative. The Lifelines initiative has been made possible by subsidy from the Dutch Ministry of Health, Welfare and Sport, the Dutch Ministry of Economic Affairs, the University Medical Center Groningen (UMCG), Groningen University and the Provinces in the North of the Netherlands (Drenthe, Friesland, Groningen).

**Competing interests:** The authors have declared that no competing interests exist.

## Conclusions

Having a contact with someone tested positive for SARS-COV-2 and living in a household with someone else are the most important factors related to a positive SARS-COV-2 test. The loss of smell or taste is the most prominent symptom associated with a positive test. Symptoms like runny nose, pain when breathing, sore throat are more likely to be indicative of other conditions.

## Background

Coronavirus disease 2019 (COVID-19), also known as the coronavirus, is an ongoing pandemic disease. Severe acute respiratory syndrome coronavirus 2 (SARS-COV-2) infection can affect anyone, and the disease symptoms range from mild to very severe. Risk factors for SARS-COV-2 severe outcomes in hospitalized patients are older age, male sex, overweight, and comorbidities such as hypertension, chronic lung disease, diabetes and coronary heart disease [1–3].

In many countries it is highly recommended to undergo SARS-COV-2 polymerase chain reaction (PCR) testing if symptoms are indicative of COVID-19 or in case of having had contact with a person who has had a positive SARS-COV-2 test. The prevalence of positive tests was about 3–4% in Europe and the US in the first year of the pandemic [4]. This indicates that the majority of the people who had an indication for COVID-19-testing were not infected by the virus. To improve the indication for getting tested some researchers investigated risk factors for having a positive test. Many published studies on risk factors of SARS-COV-2 focus on in-hospital cohorts [1]. Several population-based studies were published about the risk factors for COVID-19 diagnosis among tested individuals [5–8]. A study in Germany in symptomatic patients tested for COVID-19 found that male sex, older age, cardiac arrhythmias, depression, and obesity were positively associated with a positive SARS-COV-2 test [5]. Another study in Sweden showed that female sex, younger age and presence of a comorbidity are risk factors for having a positive SARS-COV-2 test [6]. A study in Denmark showed that subjects living with someone who tested positive for SARS-COV-2, subjects working in health care and subjects experiencing typical symptoms were more likely to have a positive SARS-COV-2 test [7]. The studies from Germany and Sweden were both focused on clinical and demographic data only and did not consider symptoms, physical activity and quality of life, while the study from Denmark evaluated the associations in univariate analyses or with a simplified model that adjusted only for a limited set of characteristics (sex, age, and household size).

In the Netherlands, with a population of 17.44 million inhabitants, more than 1.67 million confirmed cases of SARS-CoV-2 infection and about 6 thousand COVID-19 related deaths were reported until June 2021 [9]. Testing for SARS-CoV-2 by PCR was established in March 2020. Since March 2020, individuals with moderate to severe symptoms of respiratory tract infection were offered testing for SARS-CoV-2. Since April 2020, testing was available for emergency staff and individuals with mild symptoms and asymptomatic contacts, and since June 2020, nationwide, PCR-testing for SARS-CoV-2 infection became available for everyone [10]. Vaccination against COVID-19 started in January, 2021, with residents and employees of nursing homes and frontline staff at hospitals. The vaccination campaign, which started at rather low speed, was intensified in April 2021, and in August 2021 all Dutch adults had been invited for vaccination.

To identify factors for SARS-CoV-2 and address the medical, social and psychological impacts of the pandemic, a multidisciplinary group of researchers of the University Medical Center Groningen (UMCG) rapidly developed an extensive COVID-19 questionnaire and implemented this questionnaire in the already established Lifelines cohort study [11], leading to the development of the Lifelines COVID-19 cohort [12]. (Bi-)Weekly questionnaires were sent to participants to get an overview of SARS-COV-2 symptoms and related burden of this pandemic disease.

The aim of this study was to identify clinical, demographic, lifestyle and quality of life factors associated with a positive SARS-COV-2 PCR-test in subjects that were tested. Moreover, the secondary aim of this study was to determine the sensitivity and specificity of the model including the identified factors in terms of predicting a positive SARS-COV-2 PCR-test. It should be noted that the investigated factors may not have a causal relationship with a positive SARS-COV-2 PCR test, but could lead to a better defined indication for SARS-COV-2 PCR-testing and, as a result, reduce the number of people requiring a test. To optimize the identification of factors, the performance of several machine learning methods was compared.

## Methods

A population-based case-control study was conducted using data from the Lifelines COVID-19 cohort, in which all participants who completed at least one COVID-19 questionnaire were included. In the study the participants were included if they had a PCR-test for SARS-COV-2 in the period between June 2020 (since testing was available to all inhabitants of the Netherlands from this date) and April 2021 (since the vaccination-program was intensified from this date). Questionnaires were weekly or bi-weekly sent to all participants. Lifelines is a multi-disciplinary prospective population-based cohort study examining in a unique three-generation design the health and health-related behaviours of 167,729 persons living in the North of the Netherlands. It employs a broad range of investigative procedures in assessing the biomedical, socio-demographic, behavioural, physical and psychological factors which contribute to the health and disease of the general population, with a special focus on multi-morbidity and complex genetics. [11]. Lifelines is conducted according to the Declaration of Helsinki and approved by the medical ethics committee of the Universitair Medical Center Groningen (UMCG) (no. 2007/152) and is ISO certified (9001:2008 Healthcare). A written informed consent was collected from all participants.

### Outcome

The outcome was a self-reported result of a polymerase chain reaction (PCR) test for SARS-COV-2 as assessed in the questionnaire. Cases comprised individuals who tested positive for SARS-COV-2, while controls were individuals who tested negative for the infection. The first positive test within the study period was selected as the case for each participant. Controls were then matched to cases based on sex, age and the timing of their SARS-COV-2 tests, maintaining a 1:4 ratio or larger.

**Factors of interest.** The data about factors of interest were collected using a detailed questionnaire about the participant's physical and mental health and experiences on a (bi-)weekly basis [12]. The following data were collected from the questionnaire in which the first positive SARS-COV-2 test result, or in case of no positive test, the matched negative test result was reported: age, sex, recent contact with someone who tested positive for SARS-COV-2, household composition (1 or >1 household members), smoking status, BMI, fatigue, symptoms and medications used. In addition, data on quality of life, physical activity, and alcohol consumption were extracted from the questionnaire preceding the one that indicated the first PCR test

for SARS-COV-2 taken ever or the first positive test in case of multiple tests taken over time. In case of missing values these data were extracted from the same questionnaire where the data on outcome were extracted.

Age was treated as continuous variable. Smoking status was categorized into three categories: never smokers, former smokers, and current smokers. BMI was calculated from self-reported weight and height and categorized into normal ($<$ 25), overweight (between 25 and 30) and obese ($>$ 30). Fatigue was measured on a 7-point scale and categorized into a binary variable: poor (1–3) and satisfactory/good (4–7). The included symptoms are listed in Table 1. Each symptom was measured on a 5-point scale ("not at all", "a little bit", "somewhat", "quite a lot", "very much") and was categorized into a binary variable, where "not at all" and "a little bit" were categorized as "No Symptom" and the rest as "Present symptom". Self-reported use of the following medications was reported: antihypertensives, inhaler, corticosteroids in tablets, cholesterol-lowering drugs, anti-diabetic drugs, cough and pain medication. Quality of life was measured on a scale from 1 to 9 and categorized into three categories (cutoffs for categories were 1, 6 and 8) and treated as an ordinal variable. Physical activity was measured into five categories in the questionnaire based on a number of minutes spent on physical activity in the last 14 days ($<$ 100, 100–200, 200–300, 300–360, $>$360) and treated as ordinal variable. Alcohol consumption was reported in units per week and categorized into three categories (cutoffs for categories were 0, 1 and 8) and treated as categorical variable. The alcohol was treated as categorical variable, because there is conflicting evidence about the role of mild and heavy alcohol consumption on the risk of SARS-COV-2 infection [13].

## Analysis

Controls were then matched to cases based on sex, age, and the timing of their SARS-COV-2 tests, while maintaining a ratio of 1:4 or larger. The optimal matching procedure from the R package 'ccoptimalmatch' was applied.

Multivariate logistic regression was used to assess the associations between the risk factors and a SARS-COV-2 positive test. The variables in the logistic model were selected by minimizing the Akaike's Information Criterion (AIC) of the model ('glmStepAIC' method of the 'caret' package) using backwards selection method.

The performance of the logistic regression models was compared to three different nonlinear algorithms. The first one was a random forest, in which three hyperparameters were tuned: (1) number of trees starting in a range 100–500; (2) the number of variables selected for each split (mtry) was set to 2–8 in each split; and (3) the minimum node size to 1–8. The second algorithm was a support vector machine, in which the values of the hyperparameter "C" (cost of constraint) were set for a search (0.1 to 2). The third one was a gradient boosting machine, and the following hyperparameters were tuned: (i) eta 0.3, 0.5; (ii) gamma (0, 0.01); and (iii) max depth (1, 4, 6). The AUC was used as a performance metric.

The data were imbalanced with 2,564 patients testing positive for SARS-COV-2 matched to 10,225 patients with negative test results (Fig 1). This can lead to bias in the performance of machine-learning algorithms [14]. To account for this, the participants with a negative SARS-COV-2 test were randomly grouped into 4 equal subsets. For each subset, a supervised binary classification was performed between the participants with a positive and those with a negative SARS-COV-2 test using 10-fold cross-validation. 80% of the data was used for training and 20% for testing.

The number of missing values was 12% for the variable reflecting having had contact with another person tested positive for SARS-COV-2 test, 20% for alcohol consumption, 9% for physical activity and 6% for smoking (Table 1). In other variables there were occasional

**Table 1. Baseline characteristics of participants stratified into SARS-COV-2 positive and negative.**

| Variables | Number of participants with values | SARS-CoV-2- positive | SARS-CoV-2- negative | p-Value |
|---|---|---|---|---|
| Participants (%) | 12790 | 2564 (12%) | 10226 (88%) | |
| Age, mean (SD) | 12790 | 53.5 (11.7) | 53.6 (12.3) | 0.001 |
| Sex, females (%) | 12790 | 1668 (65.0%) | 6664 (65.0%) | 0.9 |
| Contact with someone positive for SARS-COV-2 | 11193 | 1577 (61.5%) | 533 (5.2%) | <0.001 |
| > = 1 household members | 12711 | 2289 (89.3%) | 8908 (87.1%) | 0.01 |
| BMI | 12392 | | | |
| Normal (%) | 5293 | 948 (37.0%) | 4345 (42.5%) | 0.03 |
| Overweight (%) | 5094 | 1070 (41.7%) | 4024 (39.4%) | |
| Obese (%) | 2005 | 463 (18.1%) | 1542 (15.1%) | |
| Missing (%) | 398 | 83 (3.2%) | 315 (3.1%) | |
| Smoking status | 12032 | | | |
| • Never (%) | 5613 | 1145 (44.7%) | 4468 (43.7)% | 0.004 |
| • Former (%) | 5147 | 1064 (41.5%) | 4083 (40.0%) | |
| • Current (%) | 1272 | 206 (8.0%) | 1066 (10.4%) | |
| • Missing (%) | 758 | 149 (5.8%) | 609 (6.0%) | |
| Alcohol consumption (categories) | 10747 | | | |
| • Low (%) | 3790 | 897 (35.0%) | 2893 (28.3) | <0.001 |
| • Moderate (%) | 3769 | 757 (30.0%) | 3012 (29.5%) | |
| • High (%) | 3188 | 586 (22.9%) | 2602 (25.0%) | |
| • Missing (%) | 2043 | 324 (12.6%) | 2036 (16.8%) | |
| Physical activity (min in 14 days) | | | | |
| • Less than 100 minutes (%) | 1540 | 514 (20.1%) | 1026 (10.0%) | <0.001 |
| • 100 to 200 minutes (%) | 2042 | 433 (16.9%) | 1609 (15.7%) | |
| • 200 to 300 minutes (%) | 1516 | 290 (11.3%) | 1226 12.0%) | |
| • 300 to 360 minutes (%) | 4599 | 775 (30.2%) | 3824 (37.4%) | |
| • More than 360 minutes (%) | 1915 | 361 (14.1%) | 1554 (15.2%) | |
| • Missing (%) | 1178 | 191 (7.5%) | 987 (9.7%) | |
| Fatigue (poor vs satisfactory/good)) | 12737 | 1528 (60.0%) | 2586 (25.3%) | <0.001 |
| Quality of life in 14 days (categories) | | | | |
| • Low (%) | 1064 | 1272 (49.6%)% | 2924 (28.6%) | <0.001 |
| • Moderate (%) | 12841 | 1079 (42.1%) | 6243 (61.0%) | |
| • High (%) | 7332 | 174 (6.8%) | 936 (9.2%) | |
| • Missing (%) | 727 | 39 (1.5%) | 132 (1.3%) | |
| Symptoms in last 14 days | | | | |
| • sensitive skin (%) | 12722 | 280 (10.9%) | 390 (3.8%) | <0.001 |
| • pain in the neck, shoulder(s) or arm(s) (%) | 12726 | 623 (24.3%) | 1385 (13.5%) | <0.001 |
| • pain in the upper back (%) | 12726 | 432 (16.8%) | 361 (3.5%) | <0.001 |
| • shortness of breath (%) | 12735 | 424 (16.5%) | 822 (4.2%) | <0.001 |
| • pain when breathing (%) | 12736 | 158 (6.2%) | 94 (1.0%) | <0.001 |
| • runny nose (%) | 12742 | 531 (20.7%) | 1420 (13.9%) | <0.001 |
| • sore throat (%) | 12737 | 430 (16.8%) | 841 (8.2%) | <0.001 |
| • dry cough (%) | 12726 | 606 (23.6%) | 650 (6.4%) | <0.001 |
| • wet cough (%) | 12725 | 352 (13.7%) | 539 (5.3%) | <0.001 |
| • fever (38 degrees or higher) (%) | 12726 | 445 (17.4%) | 167 (1.6%) | <0.001 |
| • diarrhea (%) | 12726 | 227 (8.9%) | 247 (2.4%) | <0.001 |
| • stomach pain (%) | 12728 | 175 (6.8%) | 374 (3.7%) | <0.001 |

*(Continued)*

**Table 1.** (Continued)

| Variables | Number of participants with values | SARS-CoV-2- positive | SARS-CoV-2- negative | p-Value |
|---|---|---|---|---|
| • loss of sense of smell or taste (%) | 12730 | 921 (36.2%) | 121 (1.2%) | <0.001 |
| • red, painful or itchy eyes (%) | 12730 | 278 (10.8%) | 407 (4.0%) | <0.001 |
| • sneezing (%) | 12734 | 444 (17.3%) | 1041 (10.2%) | <0.001 |
| • headache (%) | 12738 | 873 (34.0%) | 1091 (10.7%) | <0.001 |
| • dizziness (%) | 12736 | 337 (13.1%) | 279 (2.7%) | <0.001 |
| • heart or chest pain (%) | 12728 | 263 (10.3%) | 221 (2.2%) | <0.001 |
| • lower back pain (%) | 12732 | 490 (19.1%) | 1176 (11.5%) | <0.001 |
| • nausea or upset stomach (%) | 12734 | 349 (13.6%) | 495 (4.8%) | <0.001 |
| • muscle pain/aches (%) | 12734 | 942 (36.7%) | 1264 (12.4%) | <0.001 |
| • difficulty breathing (%) | 12730 | 317 (12.4%) | 214 (2.1%) | <0.001 |
| • feeling suddenly warm, then suddenly cold (%) | 12735 | 789 (30.8%) | 666 (6.5%) | <0.001 |
| • numbness or tingling somewhere in body (%) | 12737 | 201 (7.8%) | 412 (4.0%) | <0.001 |
| • lump in throat (%) | 12734 | 276 (10.8%) | 324 (3.2%) | <0.001 |
| • part of body feeling limp or heavy (%) | 12734 | 975 (38.0%) | 670 (6.6%) | <0.001 |
| • feeling of heaviness in your arms or legs (%) | 12734 | 732 (28.5%) | 488 (4.8%) | <0.001 |
| Medications | | | | |
| • antihypertensives (%) | 1582 | 326 (12.7%) | 1256 (12.3%) | 0.58 |
| • inhaler (%) | 860 | 184 (7.2%) | 676 (6.6%) | 0.33 |
| • corticosteroids in tablet form (such as prednisone) (%) | 83 | 16 (0.6%) | 67 (0.7%) | 0.97 |
| • cholesterol lowering medication (%) | 965 | 203 (7.9%) | 762 (7.5%) | 0.45 |
| • diabetes medication (%) | 266 | 60 (2.3%) | 206 (2.0%) | 0.34 |
| • cough medication (%) | 245 | 49 (1.9%) | 196 (1.9%) | 0.99 |
| • strong pain medication (%) | 195 | 44 (174%) | 151 (1.5%) | 0.43 |

missing values in about 1–4%. People with a positive SARS-COV-2 test were less likely to have missing data on symptoms. All the missing values were imputed per subset using multiple imputation by chained equation [15]. A total of five imputed datasets were generated per subset, which were then pooled for the analysis.

The overall performance of the models reported as AUC and its confidence interval (CIs). Sensitivity and specificity reported to provide information about the model's ability to correctly identify positive and negative cases and discriminate between them.

All analyses were performed in R Statistics (Version 4.0.3) with the 'caret' package.

## Results

In total, 21,964 of 71,992 participants who completed at least one COVID-19 questionnaire reported that they were tested between June 2020 and April 2021, of which 2,564 had a positive SARS-COV-2 test (Table 1). After matching by sex, age and the timing of the SARS-COV-2 tests, 10,226 controls were identified. In univariate group comparisons, subjects with a positive test were more likely to have been in contact with another person who tested positive for SARS-COV-2, were more likely to live in a household with someone else and be overweight, were less likely to be current smokers, drink less alcohol, were less physically active, had worse quality of life, more likely to have poor fatigue and exhibit typical symptoms of SARS-COV-2 more often as compared to subjects with a negative SARS-COV-2 test.

The results of multivariate logistic regression are presented in Fig 2. The variables showing a significant difference in the univariate analyses also had a significant association with a

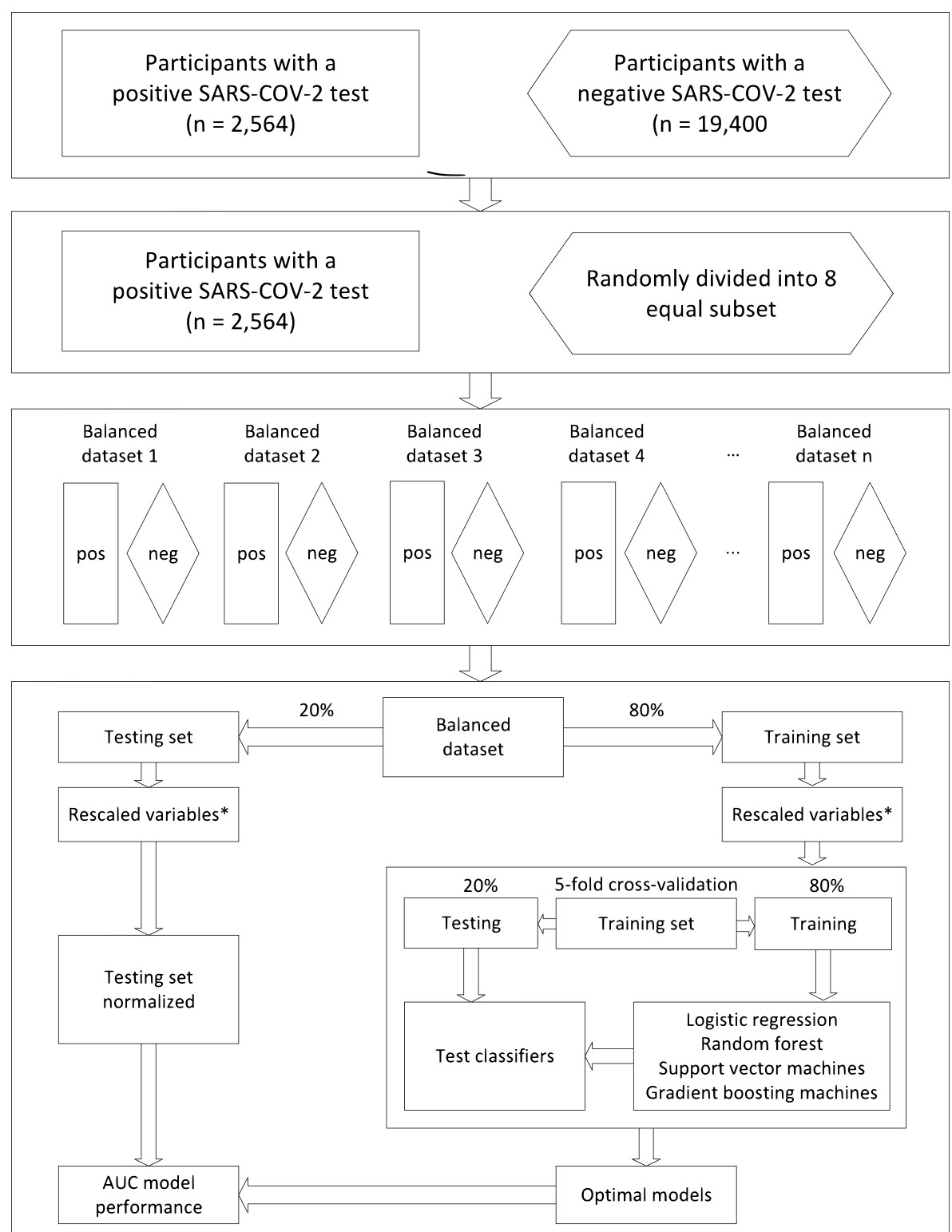

**Fig 1. Overview of the procedure followed to reduce class imbalance (equalization strategy).** * Continuous variables were rescaled to the interval between zero and one; AUC–Area Under the Curve.

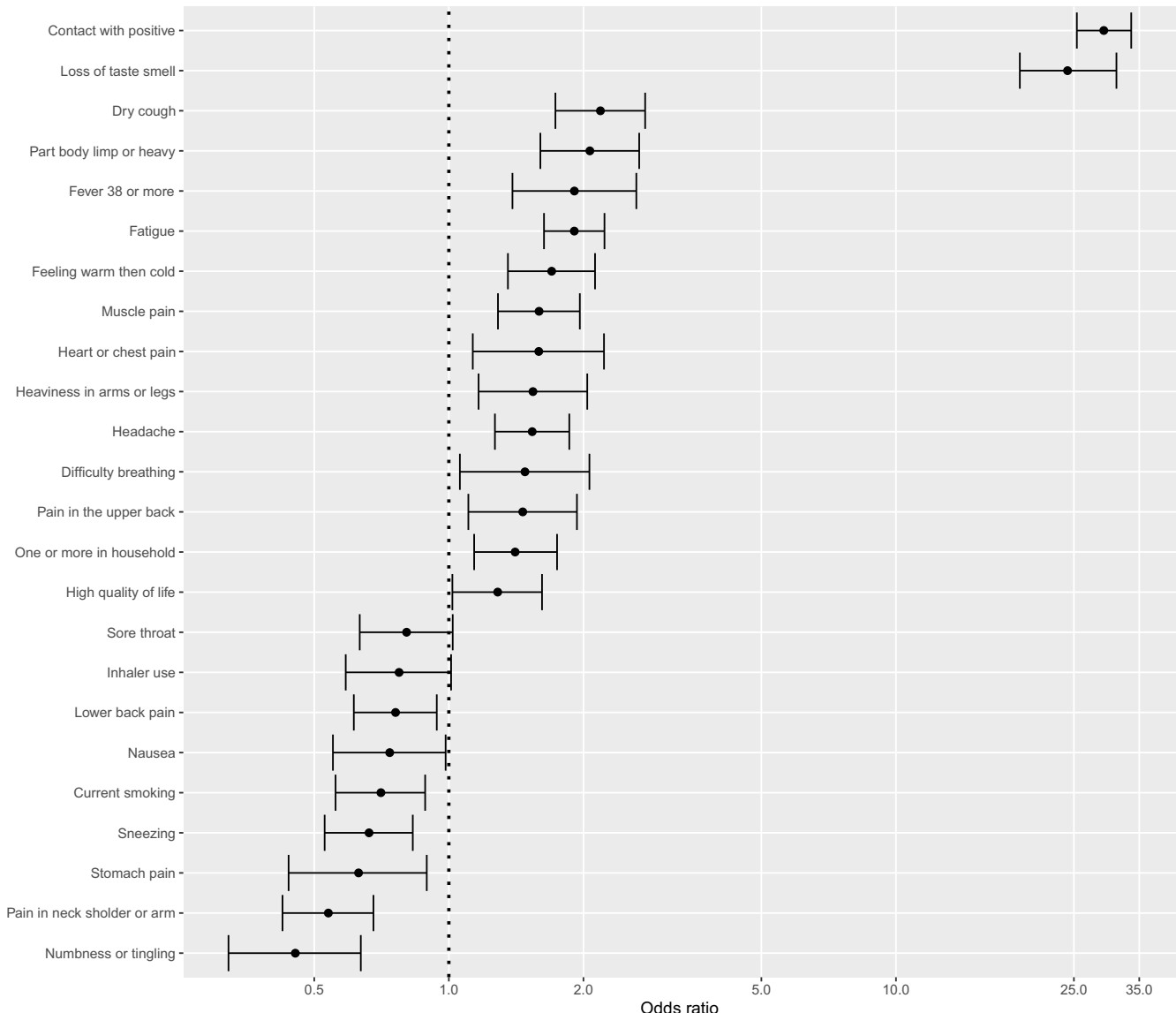

**Fig 2. Logistic regression representing the associations between the characteristics of participants and test outcome for SARS-COV-2 infection.**

SARS-COV-2 positive test in the multivariate logistic regression, i.e. contact with another person tested positive for SARS-COV-2, living in a household with someone else, and the following symptoms: loss of taste and smell, dry cough, part of body feeling lump or heavy, fever, feeling warm and then suddenly cold again, muscle pain, heart of chest pain, heaviness in arms or legs, headache, difficulty breathing, and pain in upper back (see Fig 2 for details). Of other factors, poor fatigue and higher quality of life were associated with a higher risk of a positive SARS-COV-2 test. The symptoms associated with a lower risk of a SARS-COV-2 positive test were lower back pain, nausea, sneezing, stomach pain, pain in neck, shoulder or arm and feeling of numbness or tingling in the body. Of other factors, use of an inhaler as well as being a current smoker were associated with a lower risk of a SARS-COV-2 positive test.

With a sensitivity of 88.4% the model can identify people with a positive SARS-COV-2 test and with a specificity of 81.1% the model can identify the participants with a negative

**Table 2. Overall performance of machine learning algorithms by AUCs for the 8 subsets.**

| Analysis | AUC and CIs |
|---|---|
| Logistic regression | 0.858 (0.857–0.859) |
| Random Forest | 0.860 (0.859–0.861) |
| Support Vector Machine | 0.853 (0.850–0.855) |
| Gradient Boosting Machines | 0.861 (0.860–0.862) |

SARS-COV-2 test. The overall performance of the logistic regression models was good (AUC = 0.86). The performance of the logistic regression model was comparable with the performance of random forest, support vector machine and gradient boosting machine models (Tables 2 and 3).

A sensitivity analysis was conducted restricting the cohort to people who had not been in contact with another person tested positive for SARS-COV-2 test, because that variable was dominant in the analysis and had a very large effect size (OR = 30). The findings were similar to the original model, but also showed that more symptoms not typical for SARS-COV-2 infection, like runny nose and sore throat, as well as inhalers use and higher physical activity, are associated with a lower risk of a positive SARS-COV-2 test (S1 Fig).

## Discussion

Having had contact with another person tested positive for SARS-COV-2 test and living in a household with someone else were the most important factors related to a positive SARS-COV-2 test. The most prominent symptom associated with a positive test was loss of smell or taste. Other symptoms related to SARS-COV-2 positive test were those typically observed in the course of the disease: chest pain, fever, part of body feeling lump or heavy, difficulty breathing, dry cough, feeling warm and then suddenly cold again, pain in upper back, muscle pain, heaviness in arms or legs and headache. Interestingly, symptoms that likely indicate other conditions rather than SARS-COV-2 were runny nose, lower back pain, diarrhea, sore throat, pain in neck, shoulder or arm, feeling of numbness or tingling in the body and stomach pain. Of behavioural characteristics, having poor fatigue and higher quality of life were related to a higher risk of being positively tested for SARS-COV-2. On the other hand, more intense physical activity, inhaler use, as well as being a current smoker were associated with a lower risk of a SARS-COV-2 positive test. The model including the identified factors can identify the participants with a positive SARS-COV-2 test with a high sensitivity of 88.4% and the participants with a negative SARS-COV-2 test with a moderate specificity of 81.1%.

As expected, the important factors related to a positive SARS-COV-2 test were having had

**Table 3. Proportion of predicted values averaged for 4 subsets.**

| Analysis type | | True Negative | True Positive |
|---|---|---|---|
| Logistic regression | Predicted Negative | **456 (45%)** | 105 (10%) |
| | Predicted Positive | 53 (5%) | **407 (40%)** |
| Random Forest | Predicted Negative | **446 (42%)** | 93 (9%) |
| | Predicted Positive | 63 (6%) | **419 (41%)** |
| Support Vector Machine | Predicted Negative | **450 (44%)** | 119 (12%) |
| | Predicted Positive | 59 (6%) | **393 (38%)** |
| Gradient Boosting Machines | Predicted Negative | **457 (45%)** | 104 (10%) |
| | Predicted Positive | 52 (5%) | **408 (40%)** |

contact with another person tested positive for SARS-COV-2 and living in a household with someone else. This can be explained by the transmission nature of the disease, which is mainly transmitted when people breathe air contaminated by droplets and small airborne particles containing the virus [16]. If someone has been in close contact with other people, especially with those with a recently confirmed SARS-COV-2 infection, the risk that the virus gets transmitted increases. These factors were also previously found in relation to SARS-COV-2 in other studies [7, 17].

Interesting to note, that some of the measured symptoms were inversely related to the risk of being positively tested for SARS-COV-2, such as runny nose, lower back pain, sore throat, pain in neck, shoulder or arm, feeling of numbness or tingling in the body and stomach pain. These symptoms are not commonly observed in patients with a confirmed diagnosis of the disease [17] and are likely to be related to other conditions, rather than SARS-COV-2. The observed relation between the typical SARS-COV-2 symptoms, such as loss of smell and taste, chest pain, fever, dry cough and others and the higher risk of being positively tested for SARS-COV-2 was not surprising and confirmed the associations previously observed in other studies [7, 18].

Of other investigated characteristics, this study found that poor fatigue was related to SARS-COV-2 since it could be an early sign of the disease [19]. Furthermore, higher quality of life showed to be related to a lower risk of SARS-COV-2 infection. People with lower quality of life are more likely to have lower income and being at their workplace during the pandemic [20, 21], which, in turn, are related to a higher risk of SARS-COV-2 infection. The relation between the use of inhaler and a lower risk of SARS-COV-2 infection observed in this study was not previously investigated. It has been previously shown that patients with asthma may be at lower risk of the COVID-19 related hospitalizations [22], that can be explain by their cautious behavior. Such people are being more careful concerning anti-coronavirus measures, such as social distancing and use of face masks. Finally, current smoking showed to be associated with a lower risk of SARS-COV-2 infection, which has also been shown in other studies [23, 24]. It has been proposed that nicotine may be responsible by binding to ACE2 protein, which is also a target for SARS-CoV-2 [25, 26]. However, it is important to mention that a history of smoking is associated with more severe COVID19 [27].

A strong point of this study is that it is population-based and the data were collected in a prospective way. The response rate for the questionnaire was decreasing over time, and ranged from 33 to 39% [12], which is considered relatively high. Furthermore, the results of the multi-variate logistic regression model that was used to select the most significant were compared to other supervised machine-learning algorithms and showed comparable performance. In addition, the performance was evaluated while eliminating the class-imbalance issue between the groups with positive and negative tests thus overcoming the biased accuracy.

In conclusion, people who have been in contact with another positive person or those with loss of smell and taste have a much higher risk of being tested positive for SARS-COV-2 infection. Contrary, symptoms like numbness or tingling in the body, stomach pain, sneezing, sore throat, pain in upper shoulder or arm and lower back pain are more likely to be indicative of other conditions, rather than SARS-COV-2 infection.

## Perspectives

Although the pandemic is getting under control in 2022 and the currently prevalent 'omicron' variant of SARS-COV-2 is less dangerous than 'alpha' or 'delta', the pandemic gives a lesson to learn for possible future pandemics or for more dangerous variants of the disease. Though it is not clear yet if SARS-COV-2 will become an endemic disease, the conditions and risk factors

affecting the risk of being infected should be identified in detail. This would lead to effective and efficient surveillance of risk factors and will enable a prompt response with regard to adequate testing policy in case of a new epidemic or a new wave of SARS-COV-2. The key pieces of data should be readily collected in order to make good decisions. This would be especially of importance in countries with limited options for testing, but also relevant in many developed countries, where people sometimes had to wait for more than 48 hours to get tested.

## Limitations

It is important to note that the data were self-reported, which can lead to overreporting of symptoms related to SARS-COV-2 due to increased media attention, such as loss of sense of smell and cough [28]. Though there was a small difference between the non-responders (29%) and responders (71%) in age, sex, BMI, and smoking status [12], it is not expected to impact the findings of this study. Furthermore, this study looks at the period when the 'alpha' variant of SARS-COV-2 was prevalent in the Netherlands and self-tests were not available at that time, making the findings less relevant for the currently prevalent variant of the virus. The probability of a positive result of a PCR test for SARS-COV-2 is expected to be higher within 14 days after vaccination due to immunosuppression. Patients who experience symptoms within 14 days before a positive PCR test can be expected to have lower vaccination rates, therefore, the effect size of the association between the presence of symptoms and a positive SARS-COV-2 test can be underestimated. However, in our study we only used the data until April 2021. At that time only a minority of the population was vaccinated.

## Supporting information

**S1 Fig. Logistic regression representing the associations between the characteristics of participants and test for SARS-COV-2 among people without having contact with someone tested positive within the last 14 days.**
(EPS)

## Acknowledgments

We express our sincere gratitude to the Lifelines Coronavirus Initiative whose invaluable support made the collection of study data possible, specifically to H. Marike Boezen, Jochen O. Mierau, H. Lude Franke, Jackie Dekens, Patrick Deelen, Pauline Lanting, Judith M. Vonk, Ilja Nolte, Anil P.S. Ori, Annique Claringbould, Floranne Boulogne, Marjolein X.L. Dijkema, Henry H. Wiersma, Robert Warmerdam, Soesma A. Jankipersadsing, Irene van Blokland, Geertruida H. de Bock, Judith GM Rosmalen, Cisca Wijmenga. Lifelines Coronavirus Initiative is a joint initiative of the University Medical Center Groningen, the University of Groningen, the Aletta Jacobs School of Public Health and the Lifelines biobank.

## Author Contributions

**Conceptualization:** Grigory Sidorenkov, Judith M. Vonk, Marco Grzegorczyk, Francisco O. Cortés-Ibañez, Geertruida H. de Bock.

**Data curation:** Grigory Sidorenkov, Judith M. Vonk, Francisco O. Cortés-Ibañez.

**Formal analysis:** Grigory Sidorenkov, Marco Grzegorczyk, Francisco O. Cortés-Ibañez.

**Funding acquisition:** Geertruida H. de Bock.

**Investigation:** Grigory Sidorenkov.

**Methodology:** Grigory Sidorenkov, Judith M. Vonk, Marco Grzegorczyk, Francisco O. Cortés-Ibañez, Geertruida H. de Bock.

**Project administration:** Grigory Sidorenkov.

**Resources:** Geertruida H. de Bock.

**Software:** Grigory Sidorenkov, Geertruida H. de Bock.

**Supervision:** Geertruida H. de Bock.

**Validation:** Grigory Sidorenkov, Judith M. Vonk, Marco Grzegorczyk, Geertruida H. de Bock.

**Visualization:** Grigory Sidorenkov.

**Writing – original draft:** Grigory Sidorenkov, Judith M. Vonk, Marco Grzegorczyk, Francisco O. Cortés-Ibañez, Geertruida H. de Bock.

**Writing – review & editing:** Grigory Sidorenkov, Judith M. Vonk, Marco Grzegorczyk, Francisco O. Cortés-Ibañez, Geertruida H. de Bock.

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
