## [Decision Letter · Decision Letter 0]

15 Feb 2023

PONE-D-22-29571

Factors associated with SARS-COV-2 positive test in Lifelines

PLOS ONE

Dear Dr. Sidorenkov,

Thank you for submitting your manuscript to PLOS ONE. After careful consideration, we feel that it has merit but does not fully meet PLOS ONE’s publication criteria as it currently stands. Therefore, we invite you to submit a revised version of the manuscript that addresses the points raised during the review process.

We look forward to receiving your revised manuscript.

Kind regards,

Mohamed Lounis

Academic Editor

PLOS ONE

Journal Requirements:

Reviewers' comments:

Reviewer's Responses to Questions

**Comments to the Author**

1. Is the manuscript technically sound, and do the data support the conclusions?

Reviewer #1: Partly

Reviewer #2: Partly

2. Has the statistical analysis been performed appropriately and rigorously? 

Reviewer #1: I Don't Know

Reviewer #2: No

3. Have the authors made all data underlying the findings in their manuscript fully available?

Reviewer #1: Yes

Reviewer #2: No

4. Is the manuscript presented in an intelligible fashion and written in standard English?

Reviewer #1: Yes

Reviewer #2: Yes

5. Review Comments to the Author

Reviewer #1: 1. Abstract: 2564(12%) or 21964 is not of ?

2. Perspective or Limitation needs to be commented on the association with genetic vaccination. Differences by vaccination or not, by number of vaccinations, and the point that PCR positivity is higher within 14 days after vaccination due to immunosuppression.

3. Since the Omicron strain, viral infection has been localized (nose and throat) and sore throat has become more pronounced.

4. The report only that smoking is less likely to be PCR positive misleads the reader.

It should be added that a history of smoking is associated with more severe COVID19.

Reviewer #2: This paper studies factors associated with SARS-COV2 positive tests in a cohort from The Netherlands, which was examined using questionnaires. Below are major and minor comments to the paper.

Major comments

- The overall aim of the paper appears unclear to me. In the title of the paper, focus is on "factors associated with...", whereas the paper speaks about risk factors for a positive test (l1, p4). Authors should clarify if it is merely associations or risk (causal?) they are concerned with, and if "risk factors" is the choice, it should be clarified how a symptom of Covid-19 can be a risk factor (for a positive test)?

- The previous item also has implications for the intended use of the results, which I find unclear. It seems that authors suggest their finding could be used to create some sort of high risk screening strategy instead of the broad population-based screening used in many countries in an effort to combat Covid-19. If such a high risk screening strategy is indeed the objective, authors should have focused much more on estimating the usual quality measures of a screening test such as sensitivity, specificity, positive and negative predictive values. Further I am missing considerations of the fact that since Covid-19 is a contagious disease, sensitivity or capture rates are probably much more important than specificity? If the intention is not to suggest a high-risk screening strategy, the scientific value of the paper is likely less and in any case needs to be elaborated.

- The fundamental data source is created by participants filling out a questionnaire. This raises profound questions about the validity of the self-reported data. This is only mentioned very briefly in Limitations. I fail to understand why the prevalence of a positive test is likely over-estimated with these data? I also fail to see why this in itself is a problem, if focus is on the associations with other self-reported (risk) factors? In other words, I think authors should discuss the potential for bias in these associations due to the self-reporting. Also concerning self-reported values: The first sentence of the Results state that "...21,964 patients were tested...", but would it be more precise to say that this number said they had been tested? Or is there some external data which supports this number?

- I think authors need to discuss potential bias from self-selection by invited participants - how many chose to participate and did they differ from those who did not participate? How may this have affected the results?

- I am not sure I understand the study design and why it has been used. First, the first positive PCR test for each participant is selected, if any, and then the first negative test for all other participants? This must mean that "controls" are selected conditional on not later becoming "cases", but this seems problematic from an epidemiological point of view as it conditions on the future. Perhaps it would be more straightforward to use a standard case-control study design, possibly with matching? This could also address the large variation in testing strategies over time.

- The analytic strategy is confusing in my view. The main analysis is logistic regression with automated selection of important explanatory factors. This is then compared to three other automated algorithms used in machine learning. However the first results presented (Figure 1) are estimates from logistic regression of the effect of individual factors with confidence intervals. Since an automated selection process has been used, we know a priori that only larger effects have been included and that confidence intervals are downward biased. If focus is on valid estimation of effect sizes this should be based on a pre-specified analysis plan without data-based selection of which effects to consider.

* Minor comments

p6, top: It has not yet been defined what these subsets refer to.

p6, top: How many imputed datasets were created?

p6, middle: Why should there be a balanced number of participants with a positive and negative PCR? Logistic regression estimates are unbiased by this imbalance in ordinary statistical analyses, so please explain why this should be different when labelled machine learning?

p9, top: The conclusion is overstated in my view as it gives the impression is warranted for patients with some characteristics, thereby implying implicitly that it is not warranted for others. If the objective of testing is containment of the number of infected, we need knowledge about how few it would be okay to miss with a testing strategy, before we can reach a conclusion that restricting testing to certain groups with higher risks of testing positive is appropriate. I realize that such knowledge may be largely missing, but it is nevertheless a prerequisite here.

6. PLOS authors have the option to publish the peer review history of their article (what does this mean?). If published, this will include your full peer review and any attached files.

Reviewer #1: No

Reviewer #2: **Yes: **Henrik Støvring

---

## [Author Response · Author response to Decision Letter 0]

29 Apr 2023

Dear Editor,

Thank you for the opportunity to improve our manuscript. We appreciate the given feedback and good suggestions. We have prepared the responses to the comments point by point, which you can find below. We submitted two versions of the manuscript. One was with revisions marked using tracked changes and another was a clean manuscript after revision. The revisions have been drafted in consultation with all coauthors, and each author has given approval to the final form of the revisions. 

On behalf of authors,

Kind regards,

Grigory Sidorenkov

University Medical Center Groningen, Department of Epidemiology

Groningen, the Netherlands, PO Box 30.001, FA 40, 9700 RB.

E-mail: g.sidorenkov@umcg.nl.

Tel: +31 50 361 0739 (Secretariat).

Reviewer #1: 

Comment 1. Abstract: 2564(12%) or 21964 is not of ?

 Authors’ reply: Thank you for pointing out. We have corrected the sentence.

Comment 2. Perspective or Limitation needs to be commented on the association with genetic vaccination. Differences by vaccination or not, by number of vaccinations, and the point that PCR positivity is higher within 14 days after vaccination due to immunosuppression.

Authors’ reply: Following the advice of the reviewer, we have addressed this point in the limitations and added the following text on page 9: “The probability of a positive result of a PCR test for SARS-COV-2 is expected to be higher within 14 days after vaccination due to immunosuppression. Patients who experience symptoms within 14 days before a positive PCR test can be expected to have lower vaccination rates, therefore, the association between the presence of symptoms and a positive SARS-COV-2 test can be underestimated. However, in our study we only used the data until April 2021. At that time only a minority of the population was vaccinated.”

Comment 3. Since the Omicron strain, viral infection has been localized (nose and throat) and sore throat has become more pronounced.

Authors’ reply: We have added at the limitation section on page 10 that “this study looks at the period when the ‘alpha’ variant of SARS-COV-2 was prevalent in the Netherlands and self-tests were not available at that time, making the findings less relevant for the currently prevalent variant of the virus”, which is Omicron.

Comment 4. The report only that smoking is less likely to be PCR positive misleads the reader.

It should be added that a history of smoking is associated with more severe COVID19.

Authors’ reply: We agree with the reviewer that this is an important addition. We have added the following text in the discussion section on page 10: “However, it is important to mention that a history of smoking is associated with more severe COVID19 [28].”

Reviewer #2: This paper studies factors associated with SARS-COV2 positive tests in a cohort from The Netherlands, which was examined using questionnaires. Below are major and minor comments to the paper.

Major comments

Comment 1: The overall aim of the paper appears unclear to me. In the title of the paper, focus is on "factors associated with...", whereas the paper speaks about risk factors for a positive test (l1, p4). Authors should clarify if it is merely associations or risk (causal?) they are concerned with, and if "risk factors" is the choice, it should be clarified how a symptom of Covid-19 can be a risk factor (for a positive test)?

Authors’ reply: We acknowledge that the term 'risk factors' may not be suitable for our study and have revised the manuscript accordingly. We are referring to factors that may not necessarily have a causal relationship with a positive SARS-CoV-2 test. We also made a remark about it in the introduction on page 4 adding the following sentence: “It should be noted that the investigated factors may not have a causal relationship with a positive SARS-COV-2 PCR test but could lead to a better defined indication for SARS-COV-2 PCR-testing and, as a result, reduce the number of people requiring a test”.

Comment 2: The previous item also has implications for the intended use of the results, which I find unclear. It seems that authors suggest their finding could be used to create some sort of high risk screening strategy instead of the broad population-based screening used in many countries in an effort to combat Covid-19. If such a high risk screening strategy is indeed the objective, authors should have focused much more on estimating the usual quality measures of a screening test such as sensitivity, specificity, positive and negative predictive values. Further I am missing considerations of the fact that since Covid-19 is a contagious disease, sensitivity or capture rates are probably much more important than specificity? If the intention is not to suggest a high-risk screening strategy, the scientific value of the paper is likely less and in any case needs to be elaborated.

Authors’ reply: We appreciate the reviewer for giving us this valuable advice. We have reported sensitivity and specificity and have given more attention to it in the text. 

In the introduction on page 4 we added the following text: “Moreover, the secondary aim of this study was to determine the sensitivity and specificity of the model including the identified factors in terms of predicting a positive SARS-COV-2 PCR test.“

In the methods section on page 6 we added the following text: “The overall performance of the models reported as AUC and its confidence interval (CIs). Sensitivity and specificity reported to provide information about the model's ability to correctly identify positive and negative cases and discriminate between them.”

In the results section on page 7 we added the following text: “With a sensitivity of 81% the model can identify people with a positive SARS-COV-2 test and with a specificity of 90% the model can identify the participants with a negative SARS-COV-2 test. The overall performance of the logistic regression models was good (AUC = 0.86).”

In the discussion section on page 8 we added the following text: “The model including the identified factors can identify the participants with a positive SARS-COV-2 test with a moderate sensitivity of 81% and the participants with a negative SARS-COV-2 test with a relatively high specificity of 90%.”

Comment 3: The fundamental data source is created by participants filling out a questionnaire. This raises profound questions about the validity of the self-reported data. This is only mentioned very briefly in Limitations. I fail to understand why the prevalence of a positive test is likely over-estimated with these data? I also fail to see why this in itself is a problem, if focus is on the associations with other self-reported (risk) factors? In other words, I think authors should discuss the potential for bias in these associations due to the self-reporting. Also concerning self-reported values: The first sentence of the Results state that "...21,964 patients were tested...", but would it be more precise to say that this number said they had been tested? Or is there some external data which supports this number?

Authors’ reply: We appreciate the reviewer's comment on the validity of self-reported questionnaires and acknowledge that it is difficult to determine if test positivity is reported differently than other data. In our study, we assumed that due to the media coverage and heightened attention to the topic, some individuals may have overreported a positive SARS-COV-2 PCR test. However, it has not been supported by evidence, therefore, we omit such a reporting. In the limitation we have mentioned the following: “It is important to note that the data were self-reported, which can lead to overreporting of symptoms related to SARS-COV-2 due to increased media attention, such as loss of sense of smell or taste and cough [29]”. Furthermore, we have rephrased the first sentence in the results following the remark of the reviewer: “In total, 21,964 of 71,992 participants who completed at least one COVID-19 questionnaire reported that they were tested between June 2020 and April 2021, of which 2,564 (12%) had a positive SARS-COV-2 test”. 

Comment 4: I think authors need to discuss potential bias from self-selection by invited participants - how many chose to participate and did they differ from those who did not participate? How may this have affected the results?

Authors’ reply: We have added the following sentence to the limitation section on page 9: “Though there was a small significant difference between the non-responders (29%) and responders (71%) in age, sex, BMI, and smoking status [12], it is not expected to impact the findings of this study.”

Comment 5: I am not sure I understand the study design and why it has been used. First, the first positive PCR test for each participant is selected, if any, and then the first negative test for all other participants? This must mean that "controls" are selected conditional on not later becoming "cases", but this seems problematic from an epidemiological point of view as it conditions on the future. Perhaps it would be more straightforward to use a standard case-control study design, possibly with matching? This could also address the large variation in testing strategies over time.

Authors’ reply: We acknowledge that the selection of controls conditional on their future status may be seen as a limitation of this study. However, we chose this approach to address the research question using longitudinal data. We did not select controls but consider all non-cases in a study period to be controls. We do not expect it would bias the results of this study, because the data about the SARS-COV-2 test and other factors were in most cases collected from the questionnaire at the same time point. It was common, that people received multiple tests for SARS-COV-2 infection during the study period. Using a case-control design with matching may be another way of dealing with the data, however, it could also introduce other biases, such as selection bias. 

Comment 6: The analytic strategy is confusing in my view. The main analysis is logistic regression with automated selection of important explanatory factors. This is then compared to three other automated algorithms used in machine learning. However the first results presented (Figure 1) are estimates from logistic regression of the effect of individual factors with confidence intervals. Since an automated selection process has been used, we know a priori that only larger effects have been included and that confidence intervals are downward biased. If focus is on valid estimation of effect sizes this should be based on a pre-specified analysis plan without data-based selection of which effects to consider.

Authors’ reply: We acknowledge that CIs may be slightly biased downwards due to the automatic selection of the variables, but we do not expect such bias to be relevant, since only a few variables were excluded by the automatic selection. We consider selection based on AIC as an optimal way of selecting relevant variables in our analysis. To ensure that our logistic regression model performed well, we compared its performance to three different nonlinear algorithms, which was comparable.

* Minor comments

Comment 7: p6, top: It has not yet been defined what these subsets refer to.

Authors’ reply: We thank the reviewer for pointing this out. We have shifted the sentences about the missing data and the way of dealing with it to the end of the ‘analysis’ subsection to satisfy the logical order with regard to the subsets definition.

Comment 8: p6, top: How many imputed datasets were created?

Authors’ reply: On page 6 we now clarified that “A total of five imputed datasets were generated per subset, which were then pooled for the analysis.”

Comment 9: p6, middle: Why should there be a balanced number of participants with a positive and negative PCR? Logistic regression estimates are unbiased by this imbalance in ordinary statistical analyses, so please explain why this should be different when labelled machine learning?

Authors’ reply: Class imbalance can lead to bias in machine learning because it can cause the model to favor the majority class and ignore the minority class. This results in poor performance of the model in predicting the minority class, which is the number of participants with a positive SARS-COV-2 test. In the case of class imbalance, sensitivity is more affected than specificity. This is because the minority class include fewer participants, and the model tends to have a higher false negative rate, resulting in lower sensitivity. On the other hand, specificity remains high as the majority class has more participants. Class imbalance can cause the AUC to be artificially high, which can give a false sense of model performance. Therefore, it is important to consider both sensitivity and specificity in addition to AUC when evaluating the performance of a model in the presence of class imbalance. As the number of participants with a negative SARS-COV-2 test is 8 times larger, we randomly selected 8 equal subsets from those. We have now added the reference [14] to a paper on page 6 that describes the bias potential of class imbalance in machine learning.

Comment 10: p9, top: The conclusion is overstated in my view as it gives the impression is warranted for patients with some characteristics, thereby implying implicitly that it is not warranted for others. If the objective of testing is containment of the number of infected, we need knowledge about how few it would be okay to miss with a testing strategy, before we can reach a conclusion that restricting testing to certain groups with higher risks of testing positive is appropriate. I realize that such knowledge may be largely missing, but it is nevertheless a prerequisite here.

Authors’ reply: We have rephrased the conclusion to avoid too strong wording: “In conclusion, people who have been in contact with another positive person or those with loss of smell and taste have a much higher risk of being tested positive for SARS-COV-2 infection.”

---

## [Decision Letter · Decision Letter 1]

26 May 2023

PONE-D-22-29571R1Factors associated with SARS-COV-2 positive test in LifelinesPLOS ONE

Dear Dr. Sidorenkov,

Thank you for submitting your manuscript to PLOS ONE. After careful consideration, we feel that it has merit but does not fully meet PLOS ONE’s publication criteria as it currently stands. Therefore, we invite you to submit a revised version of the manuscript that addresses the points raised during the review process.

We look forward to receiving your revised manuscript.

Kind regards,

Mohamed Lounis

Academic Editor

PLOS ONE

Additional Editor Comments:

I would thank the authors for their efforts to improve the quality of the manuscript. Howerver one of our reviewer still has a "major concen" that should be adressed

Reviewers' comments:

Reviewer's Responses to Questions

**Comments to the Author**

1. If the authors have adequately addressed your comments raised in a previous round of review and you feel that this manuscript is now acceptable for publication, you may indicate that here to bypass the “Comments to the Author” section, enter your conflict of interest statement in the “Confidential to Editor” section, and submit your "Accept" recommendation.

Reviewer #1: All comments have been addressed

Reviewer #2: (No Response)

2. Is the manuscript technically sound, and do the data support the conclusions?

Reviewer #1: Yes

Reviewer #2: Partly

3. Has the statistical analysis been performed appropriately and rigorously? 

Reviewer #1: I Don't Know

Reviewer #2: Yes

4. Have the authors made all data underlying the findings in their manuscript fully available?

Reviewer #1: Yes

Reviewer #2: No

5. Is the manuscript presented in an intelligible fashion and written in standard English?

Reviewer #1: Yes

Reviewer #2: Yes

6. Review Comments to the Author

Reviewer #1: This manuscript is well revised. It is sincere in its response to our detailed points. There are many issues regarding PCR for coronaviruses, but this study is responsive to the situation at the time.

Reviewer #2: Following the revision, the paper is now much easier to follow and in general authors have addressed the points I have raised. There is however one issue remaining, which I find crucial as it pertains to the study design. I appreciate that authors now have made the study design clear, but I think it also highlights how it is unusual and will by necessity have to influence the interpretation and conclusions of the study. The issue is with how controls have been selected.

Authors have selected controls as those never testing positive within the follow-up. I raised the concern that it might be problematic since it conditions on the future - I still think that is valid criticism, and I find authors' statement unconvincing, when they just note that this is what they chose to do (response letter, reviewer II, reply to comment 5). With the clearer description of the study design, I would like to strengthen my critique here: The participants compared are those testing positive (at least once) with those never testing positive (within the study follow-up). If authors want to keep this, I think the specificity no longer concerns an individual with true negative disease status at the time of a single test, but it is the specificity of individuals who are truly negative at the time of all PCR tests within the study. I think this is a less relevant question when considering the aim of finding factors associated with single test results. I would therefore much prefer that authors used a (matched) case-control study design to select relevant negative tests without conditioning on the future of individual patients test history.

7. PLOS authors have the option to publish the peer review history of their article (what does this mean?). If published, this will include your full peer review and any attached files.

Reviewer #1: No

Reviewer #2: **Yes: **Henrik Støvring

<quillbot-extension-portal></quillbot-extension-portal>

---

## [Author Response · Author response to Decision Letter 1]

11 Oct 2023

Review Comments to the Author

Reviewer #2: Following the revision, the paper is now much easier to follow and in general authors have addressed the points I have raised. There is however one issue remaining, which I find crucial as it pertains to the study design. I appreciate that authors now have made the study design clear, but I think it also highlights how it is unusual and will by necessity have to influence the interpretation and conclusions of the study. The issue is with how controls have been selected.

Authors have selected controls as those never testing positive within the follow-up. I raised the concern that it might be problematic since it conditions on the future - I still think that is valid criticism, and I find authors' statement unconvincing, when they just note that this is what they chose to do (response letter, reviewer II, reply to comment 5). With the clearer description of the study design, I would like to strengthen my critique here: The participants compared are those testing positive (at least once) with those never testing positive (within the study follow-up). If authors want to keep this, I think the specificity no longer concerns an individual with true negative disease status at the time of a single test, but it is the specificity of individuals who are truly negative at the time of all PCR tests within the study. I think this is a less relevant question when considering the aim of finding factors associated with single test results. I would therefore much prefer that authors used a (matched) case-control study design to select relevant negative tests without conditioning on the future of individual patients test history.

Authors reply:

We acknowledge that the selection of controls conditional on their future status may be seen as a limitation of this study. We revised the paper where necessary and used case-control study design as suggested by the reviewer. Cases comprised individuals who tested positive for SARS-COV-2, while controls were individuals who tested negative for the infection. The first positive test within the study period was selected as the case for each participant. Controls were then matched to cases based on the timing of their SARS-COV-2 tests, maintaining a 1:4 ratio. This is also reported in the methods sectionof the paper on pages 4 and 5.

The changes only slightly affected the findings. Of the specific changes, we would like to point out the changes in the magnitude of the sensitivity and specificity values, where sensitivity is increased ot relatively high (88.4) and sensitivity decreased to moderate (81.1).

All the adjustments were highlighted in the text.

---

## [Decision Letter · Decision Letter 2]

5 Nov 2023

Factors associated with SARS-COV-2 positive test in Lifelines

PONE-D-22-29571R2

Dear Dr. Sidorenkov,

We’re pleased to inform you that your manuscript has been judged scientifically suitable for publication and will be formally accepted for publication once it meets all outstanding technical requirements.

Kind regards,

Mohamed Lounis

Academic Editor

PLOS ONE

Additional Editor Comments (optional):

Reviewers' comments:

Reviewer's Responses to Questions

**Comments to the Author**

1. If the authors have adequately addressed your comments raised in a previous round of review and you feel that this manuscript is now acceptable for publication, you may indicate that here to bypass the “Comments to the Author” section, enter your conflict of interest statement in the “Confidential to Editor” section, and submit your "Accept" recommendation.

Reviewer #2: All comments have been addressed

2. Is the manuscript technically sound, and do the data support the conclusions?

Reviewer #2: Yes

3. Has the statistical analysis been performed appropriately and rigorously? 

Reviewer #2: Yes

4. Have the authors made all data underlying the findings in their manuscript fully available?

Reviewer #2: (No Response)

5. Is the manuscript presented in an intelligible fashion and written in standard English?

Reviewer #2: Yes

6. Review Comments to the Author

Reviewer #2: (No Response)

7. PLOS authors have the option to publish the peer review history of their article (what does this mean?). If published, this will include your full peer review and any attached files.

Reviewer #2: **Yes: **Henrik Støvring

---

## [Editor Report · Acceptance letter]

10 Nov 2023

PONE-D-22-29571R2 

Factors associated with SARS-COV-2 positive test in Lifelines 

Dear Dr. Sidorenkov:

I'm pleased to inform you that your manuscript has been deemed suitable for publication in PLOS ONE. Congratulations! Your manuscript is now with our production department. 

Kind regards, 

on behalf of

Dr. Mohamed Lounis 

Academic Editor

PLOS ONE